# Hematological and immunological profiles of podoconiosis patients in West Gojjam Zone, Ethiopia: A comparative cross-sectional study

Aytenew Atnaf [1,2]*, Ryan Oakes[3], Bruno T. Scodari[3,4], Kassu Desta[2], Bineyam Taye[3], Aster Tsegaye[2]

1 Department of Medical Laboratory Science, Debre Markos University, Debre Markos, Ethiopia, 2 Department of Medical Laboratory Sciences, Addis Ababa University, Addis Ababa, Ethiopia, 3 Department of Biology, Colgate University, Hamilton, New York, United States of America, 4 Department of Biomedical Data Science, Geisel School of Medicine at Dartmouth, Lebanon, New Hampshire, United States of America

* atnafnew5@gmail.com

## Abstract

### Background

Podoconiosis is a geo-chemically induced, non-infectious, familial, chronic lymphedema of the legs that occurs among barefoot people in rural, farming communities with extreme poverty. Despite a growing body of research surrounding the disease, the pathogenesis of the disease is relatively unknown. This study aims to investigate the immunological and hematological profiles of individuals affected by podoconiosis in comparison to healthy controls.

### Methodology/Principal findings

A comparative cross-sectional study was conducted in West Gojjam Zone of Ethiopia involving adult individuals clinically diagnosed with podoconiosis (n = 53) and healthy controls (n = 67) from the same area. A survey was conducted to gather information on sociodemographic, lifestyle characteristics, and clinical features of the disease. About nine ml whole blood samples were collected for hematological and immunological testing, which included IL-4, TNF-α, IL-6, IL-17, IL-10, TGF β and IFN-γ. Overall, we observed significant differences in hematological parameters between individuals with podoconiosis and healthy controls. Specifically, we found a notable reduction in white blood cell count, with an adjusted mean difference (AMD) of -1.15 (95% CI: -2.09 to -0.21; p = 0.017). Additionally, the differential white blood count showed a decrease in absolute neutrophils (AMD = -3.42, 95% CI: -4.15 to -2.69; p < 0.001) and absolute eosinophils (AMD = -0.20, 95% CI: -0.37 to -0.03; p = 0.019). Conversely, we noted an increase in absolute lymphocytes (AMD = 0.98, 95% CI: 0.50 to 1.46; p < 0.001) and monocytes (AMD = 0.54, 95% CI: 0.22 to 0.85; p = 0.001). However, we didn't observe a significant difference in cytokine profile between podoconiosis patients and healthy controls

### Conclusions/Significance

The decrease in neutrophil counts among podoconiosis cases compared to healthy controls may provide insight into the potential disease pathogenesis, suggesting the involvement of

**Data Availability Statement:** All the data are available from the Addis Ababa University repository (https://etd.aau.edu.et/server/api/core/

bitstreams/aa6b0e83-289b-45e6-b314-03375c9423eb/content).

**Funding:** We obtained financial support from the International Orthodox Christian Charity (IOCC), Ethiopia, for data collection and laboratory analyses. The Colgate University Research Council grant covered the cost of the cytokine measurements. BT received funding from the Colgate University Research Council, while AA, AT, and KD received funds from IOCC. The funders had no role in the study design, data collection and analysis, decision to publish, or preparation of the manuscript.

**Competing interests:** The authors have declared that no competing interests exist.

autoimmune-related mechanisms, as it demonstrates a similar hematological profile to other autoimmune disorders.

## Author summary

Podoconiosis, a non-infectious form of elephantiasis, is a neglected tropical disease whose immunopathology remains poorly understood. This study aimed to investigate the hematological and immunological aspects of podoconiosis. Our findings revealed significant hematological changes, particularly a reduction in absolute neutrophil counts in podoconiosis cases, which may suggest an autoimmune process. We hypothesize that chronic inflammation activates immune cells (lymphocytes, macrophages, or monocytes), which may contribute to the destruction of circulating neutrophils. We also found reduced hemoglobin levels in podoconiosis patients, possibly due to mechanisms such as hemolysis, iron sequestration, and cytokine-mediated inhibition of erythropoiesis. However, future mechanistic research is needed to confirm our observations and elucidate the role of reduced neutrophils in the pathogenesis of podoconiosis.

## Introduction

Podoconiosis, also known as endemic non-filarial elephantiasis, is considered as a geochemical disease caused by prolonged barefoot exposure to soil of volcanic origin, particularly alkali-rich volcanic soil. [1]. This preventable condition leads to severe bilateral lymphedema in the lower limbs and is commonly found among adults who have frequent contact with certain types of soil without footwear [2]. Mapping of the disease has indicated that there are endemic rates of the disease throughout highland regions of Africa, Central and South America, and northern India [3]. Despite the disease's relative preventability through shoe use and foot hygiene, it represents a major source of economic and social distress for the 1.54 million individuals estimated to be affected in Ethiopia alone [4,5].

In the past decade, podoconiosis research has accumulated a wide range of evidences surrounding the environmental risk factors as well as the host-related risk factors for the disease, especially in Ethiopia. A geo-chemical study linked specific minerals (smectite, mica, and quartz) in the soil with a higher prevalence of podoconiosis, suggesting that some feature of these non-organic soil particles may have a role in disturbing the lymphatic systems of susceptible people [6]. Further environmental studies have shown that the climate has been associated with podoconiosis risk, with people of higher altitudes, greater vegetation indices, and larger amounts of precipitation having a greater risk for disease [7]. Recently, geochemical researchers have argued that podoconiosis is associated with weathered blue soil rather than volcanic soil. Additionally blue soil indicates a greater risk of podoconiosis compared to red clay soil [8]. While these studies propose environmental risks, they fail to provide a detailed mechanism of how the geo-chemical minerals affect the lymphatic and immune systems, two of the key components of the disease pathogenesis.

Host-related characteristics significantly influence a population's susceptibility to podoconiosis. Both genetic and behavioral traits contribute to the vulnerability of these populations. A previous study indicated that genetics account for 63% of the development of podoconiosis. [9]. Additionally, a genome-wide association study linked specific polymorphisms within the HLA II cluster of genes, genes related to the production of different T-cell responses, as a

major risk factor [10]. Social behaviors and socioeconomic status have significant implications for podoconiosis. Farmers, people with uncovered floors in their homes, and those who cannot afford or choose not to wear shoes are at a high risk for developing the disease [11]. A recent hematological study indicated that while most parameters remained within normal ranges, over 40% of patients exhibited abnormalities in plateletcrit (PCT), platelet distribution width (PDW), mean platelet volume (MPV), mean cell volume (MCV), and red cell distribution width (RDW) [12].

Despite the evidence for environmental and host-related risk factors, the pathogenesis of podoconiosis is relatively unknown [13]. One previous investigation showed a potential mechanism of oxidative stress markers, which included peroxides, antioxidants, superoxide dismutase, and total growth factors, but it was still lacking the immunological data that would lend itself to a meaningful mechanism for this complex disease and its progression [14]. Another study provides a plausible hypothesis that chronic inflammation leads to lymphatic obstruction and to progressive, asymmetrical swelling of the lower legs [15]. A qualitative review compiled existing mechanisms for similar diseases to speculate on the mechanism for podoconiosis and proposed that unsuccessful clearance of silica by macrophages leads to dysregulated wound healing, the hyperproliferation of fibroblasts, thickening of lymph sites, and hard, fibrotic nodule formation [16]. A recent study in Ethiopia has shown that the pathogenesis of podoconiosis is related to over expression of genes within the pro-inflammatory axis leading to immune activation and inflammation [17]. Although each of these count as workable hypotheses, with inflammatory processes intimately linked with a broad range of chronic diseases, there is a lack of data showing the role of inflammatory cytokines and other blood cells in podoconiosis patients to justify these claims. Hence, this study aims to determine the immunological and hematological profile of individuals affected with podoconiosis compared with healthy controls. The findings of the study could shed light on how immunological markers are involved in disease progression.

## Methods

### Ethics statement

The Amhara Regional Health Bureau Research Ethics Committee and Addis Ababa University, College of Health Sciences, Department of Medical Laboratory Science Research and Ethics review committee (DRERC) approved the study with reference number HRTT/01/672/09, and DRERC 225/A/17 MLS, respectively. Written, informed consent was obtained from all study subjects. Participants were also informed about their ability to withdraw from this study at any time without jeopardizing their right to receive health services. Procedures for collection of blood samples were fully explained and were carried out by experienced professionals using sterile disposable materials. The study participants who had intestinal parasitic infections were treated in the health centers. In addition, study participants who had abnormal CBC results were linked to the health center and hospital for follow up and treatments.

### Study setting and design

A comparative cross-sectional study was conducted in a highly podoconiosis endemic area in West Gojjam zone, Ethiopia, from June 2017 –May 2018. The study focused specifically on areas within Yilmana Densa woreda, located 43 km south of Bihar Dar city and situated 2,216 m above sea level at latitude of 11˚ 29′ 60′′ North / 37˚19′ 60′′ East [18]. Patients were identified through the International Orthodox Christian Charities (IOCC) podoconiosis project in Goshiye and Debremewi Health centers in West Gojjam, enabling participants to be recruited prior to commencing treatment. The IOCC runs 11 podoconiosis clinics in the Amhara

region, one of the most heavily affected regions in Ethiopia [19]. Since its establishment in 2010, it has provided treatment to more than 10,000 patients through these clinics [19].

A trained expert from IOCC and a physician identified cases, performed staging, and collected other clinical information on patients. Then, we recruited podoconiosis patients at disease stages 1, 2, 3 and 4/5 (the highest two stages both represent advanced disease, likely to be least informative in terms of disease process), in addition to healthy controls. We screened for patients as they registered at the clinic sites and invited all those who met specific eligibility criteria. Invited study cases included diagnosed podoconiosis patients age 18 years and older who had not received prior formal lymphedema treatment. Controls were individuals 18 years and older who lived in the same areas of the cases for at least 10 years, without podoconiosis and other chronic diseases, and without a first-degree relative (parent, sibling, or child) with podoconiosis. Of the eligible invited patients, 53 cases and 67 controls were successfully recruited (n = 120).

## Sample size and sampling technique

The sample size was calculated using Epi-info software following the assumption of 90% confidence level, 80% power, 1.26 ratio of healthy controls to patients and mean and standard deviation of hemoglobin concentration of patients (13.7±2.29g/dl) and healthy controls (14.7± 2.04 g/dl) [20], respectively. Finally, a total of 53 podoconiosis patients and 67 healthy controls, a total sample size of 120. Both podoconiosis patients and healthy controls were recruited by using consecutive convenient sampling techniques.

## Measurement and data collection

After gaining written informed consent from the study participants, a predesigned interviewer-administered questionnaire was used to collect socio-demographic, and lifestyle, information regarding sex, age, monthly income, residency, religion, educational status, occupation, marital status, and shoe wearing history, and clinical information such as leg swelling history and physical examination for staging. Other information such as frequency of leg washing, shoe wearing habits, and home floor covering status was also documented. Additionally, each participant was administered a stool cup with instructions on how to provide a fecal sample. Fecal samples were placed into plastic containers with 10% formalin and transported to Addis Ababa University, Department of Medical Laboratory Sciences for parasitology analysis. To determine immuno-hematological profiles, 9 ml whole blood samples were collected from study subjects via the vacutainer method using ethylene diamine tetra-acetic acid (EDTA) tubes (4ml) and serum-separating tubes (SSTs) (5ml). EDTA anticoagulated blood samples were used to perform complete blood count and CD4+ T-cell counts. In addition, these samples were also used for ABO blood group determination and malaria microscopy by utilizing the Giemsa stain and rapid diagnostic kits. SST blood samples were centrifuged, separated into nunc tubes, stored below -20°C, and transported to Addis Ababa for cytokine level measurement.

## Hematological testing

Four ml whole blood samples were collected into EDTA tubes from each study participants and transported to Amhara Public Health Institute (APHI) laboratory, Bahir Dar, Ethiopia. Blood samples were analyzed within 3–4 hours of specimen collection using automated hematological analyzer (Cell-Dyn1800 (Abbott diagnostics, USA)). The instruments were monitored daily with normal, high and low control materials provided by the manufacturer before running the specimen to ensure quality. The analyzer aspirates the blood sample, dilutes and

counts leukocytes, erythrocytes and thrombocytes, measures MCV and hemoglobin, calculates Hct, MCH and MCHC and RDW. In addition, the instrument provides three-part differential count (Lymphocytes, Monocytes and Granulocytes; both absolute and relative counts). Blood smears were prepared from fresh whole blood and stained with Wright stain for manual differential white cell count (neutrophils, monocytes, lymphocytes, basophils, eosinophils). Absolute counts were obtained by multiplying the differential count (a percentage) by total WBC. Hemoglobin concentration adjustment was made on the altitude increment based on the WHO guideline [21].

### Flow cytometric analysis

CD4+ T-cell counts (CD4) and CD4+ percentages (%CD4) were determined by flow cytometry using FACSCount (Becton Dickinson Biosciences Immunocytometry Systems, San Jose, California, USA) and fluorochrome conjugated monoclonal antibodies. We used ready-made reagents tubes containing fluorescent-labeled anti-CD4+ monoclonal antibodies and a known number of reference beads for absolute count calculations. To ensure the quality of T-cell subset data, four levels of control beads (zero, low, medium, and high) were run on the FACSCount.

### Cytokine measurements

Cytokines are small molecules that assist the immune system in fighting infections and maintaining balance [22]. Pro-inflammatory cytokines promote inflammation, while anti-inflammatory cytokines facilitate fibrosis. In laboratory studies, cytokines help elucidate the immune system's role in diseases by indicating the nature of the immune response [23]. The serum levels of cytokines were measured by the ELISA method, using commercially available ELISA kits (ABclonal, Woburn, MA, USA) according to the manufacturer instructions. This assay employs the quantitative sandwich enzyme immunoassay technique. A monoclonal antibody specific for each cytokine (IL-4, TNF-α, IL-6, IL-17, IL-10, TGF β and IFN-γ) was pre-coated onto a microplate well. Standards and samples were pipetted into the wells and if any of the above-mentioned cytokine present in the sample, they are bounded by the immobilized antibody. Following incubation, unbound samples were removed during a wash step, and then a detection antibody specific for each cytokine was added to the wells and binds to the combination of capture antibody-cytokines in sample. Following a wash step to remove any unbound combination, an enzyme conjugate was added to the wells. After incubation and wash steps, a substrate was added. A colored product TMB was formed in proportion to the number of cytokines present in the sample. The reaction was terminated by addition of 0.1N sulfuric acid, and absorbance was measured using ELISA reader at 450 nm. Serum cytokine concentrations were calculated from a standard curve [24]. The minimum detection dose of serum cytokines was 4pg/ml for TNF-α, IFN-γ, IL-4, and IL-17 but 0.7pg/mL for IL-6, while 15 pg/mL for IL-10 and TGF β.

### Blood group determination

ABO and Rh blood grouping was done using commercially available monoclonal anti-A, anti-B, and anti-D anti-sera following methods described by Cheesbrough [25]. The rapid forward (cell) grouping method was used.

### Parasitological analysis

A drop of normal saline was added to a microscopic slide along with approximately 2 mg of stool sample. An applicator stick was used to emulsify the mixture before the slide was covered

with a cover slip. Microscopic examination for the presence of parasite eggs, larvae, trophozoite, cyst stage, and adult worm, using 10X and 40X objectives. In addition to direct wet mount examination, the formol-ether concentration technique was applied. An applicator stick was used to emulsify approximately 1 g of stool sample in 4 ml 10% formol-ether contained in a tube. Another 4 ml 10% formol-ether was added to the tube before homogenization. Emulsified feces were sieved, collected in a tube, and transferred to a centrifuge tube. Then 4 ml diethyl ether was added, and the tube was stoppered and mixed for a minute. The stopper was loosened, mixed, and centrifuged at 1000 g for 1 minute. Fecal debris was loosened and decanted, and the bottom of the tube was tapped to resuspend and mix the sediment. Sediment was placed on a slide with cover slip and examined via microscopy for ova/larvae/adult stage of the intestinal parasites.

To identify malaria parasite, both thin and thick blood films were prepared and examined microscopically after staining with 10% Giemsa for 10 minutes. The thin film was fixed with absolute methanol (96% v/v) before staining. To increase our sensitivity in detection of malaria parasite, we run Rapid Diagnostic Test (RDTs) (Malaria Pf/Pan Antigen test, Humasis, Republic of Korea) that can detect malaria antigens in a small amount of blood (5 μL) and is based on the immunochromatographic principles; capture of parasite antigen using monoclonal antibodies against a malaria antigen following the manufacturer's instruction.

## Data analysis

Data were entered, cleaned and analyzed using SPSS statistics version 25 software. In order to measure correlative associations between the disease state (either case or control) and the sociodemographic information, a chi-squared analysis of the frequencies of cases versus controls were conducted, with a p-value reported to denote significance with $p \leq 0.05$ as the standard for significance. For both hematological and immunological information, a mean value and standard deviation was calculated, with correlation compared using both t-test mean-difference and adjusted mean-difference analyses to control for covariates. The multivariate analysis occurred with a stepwise inclusion of covariates within margins of $p = 0.5$. In these cases, significance was denoted as a $p \leq 0.05$. The immunological factors were adjusted using the natural logarithm (ln) function to measure percentage change after measuring the normality and determining the logistic regression was a better fit. Further analysis looked at patient characteristics, comparing mild (1), moderate (2), and advanced (3/4/5) stages of the disease, measuring the frequencies of certain patient characteristics in combination with a chi-squared analysis. Additionally, all hematological and immunological factors were analyzed from mild to advanced stages of the disease to see if disease stage plays a role in the hematological or immunological profile of the individuals. These measures were compared using an ANOVA analysis to test for the difference between the three groups, recording the F, degree of freedom, 95% confidence interval (CI) and p-value.

## Results

### Study subjects' characteristics

A total of 120 total study subjects (53 podoconiosis patients and 67 controls) were involved in the study. Of the study population, 65.8% were male (64.2% of patients and 67.2% of controls with $p = 0.730$). The age of the patients was on average greater than the age of the controls (mean± standard deviation = 49.94±13.9 versus 41.46±13.47 years old, $p = 0.033$). The majority of both patients and controls were illiterate, but there were significantly more illiterate patients than controls (84.9% for patients and 64.2% for controls, $p = 0.011$).

The majority of both groups reported the tendency to wash their legs every day, but there were significant differences between the groups in shoe wearing history (88.7% of patients and 98.5% of controls, p = 0.023). No cases of malaria were detected using both the rapid diagnostic test (RDT) and microscopy. Overall, the burden of intestinal parasites was 35% (CI: 27–44%), with slightly higher proportion in podoconiosis patients compared with controls (37.7 vs.32.8%). ABO blood group distribution was also similar in both groups (Table 1).

## Hematological profile among podoconiosis patients and healthy controls

A univariate and multivariate analysis of the hematological variables were performed between podoconiosis patients and controls. During the analysis, patients had a significant decrease in their red blood cell (RBC) and white blood cell counts. The latter retained a significant difference even after adjustment (AMD = -1.15, 95% CI: -2.09, -0.21; p = 0.017). Additionally, absolute neutrophil (AMD = -3.42, 95% CI: -4.15, -2.69; p < 0.001) and absolute eosinophil (AMD = -0.20, 95% CI: -0.37, -0.03; p = 0.019) counts were significantly reduced in the patient group. Hemoglobin levels and red cell indices, including Mean Corpuscular Hemoglobin (MCH) and Mean Corpuscular Hemoglobin Concentration (MCHC), were significantly reduced in patients compared to healthy controls (P<0.01). We also found a significant reduction in CD4% in the podoconiosis patient compared to healthy controls (AMD = -3.39, 95% CI: -6.39, -0.40; p = 0.027). On the other hand, absolute lymphocyte and monocyte counts were significantly higher among podoconiosis patients compared to healthy controls (AMD = 0.98, 95% CI: 0.50, 1.46; p < 0.001, AMD = 0.54, 95% CI: 0.22, 0.85; p = 0.001, respectively) (Table 2).

## Immunological profile among podoconiosis patients and healthy controls

After logarithmic transformation, we compared the mean cytokine levels between podoconiosis patients and controls. Our observation showed a non-significant reduction in TNF-α (p = 0.246), Interleukin (IL)-10 (p = 0.465), and IL-6 (p = 0.972). However, we found slightly elevated measures of IL-17 (p = 0.885), IL-4 (p = 0.903), IFN-γ (p = 0.749), and TGF-β (p = 0.292) in the patient group as compared to the control group (Table 3).

## Immuno-hematolgical profile by podoconiosis clinical stages

An ANOVA between-group analysis of the hematological and immonological profiles for patients of mild, moderate, and advanced stages was performed. The immunological variables were once again tranformed using a logarithmic function to measure the percentage change for each cytokine. The hematological variables that significantly varied between staging groups were RBC (p = 0.003), WBC (p = 0.002), HGB (p < 0.001), Abs Lymphocytes (p = 0.001), Abs Monocytes (p = 0.018), Abs Neutrophil (p < 0.001), MCV (p < 0.001), and MCH (p < 0.001). Of these, RBC, HGB, Abs Lymphocyte, and MCH showed a decrease from mild to advanced staging while WBC, Abs Monocyte, Abs Neutrophil, and MCV showed an increase from mild to advanced staging. As for immunological factors, there were no significant differences between groups, but there were trends to report in terms of increasing or decreasing measurements as the stages progressed. TNF-α (p = 0.333), IL-17 (p = 0.850), and IFN-γ (p = 0.944) each increased from the mild to advanced staging, while IL-10 (p = 0.208), IL-4 (p = 0.981), IL-6 (p = 0.916), and TGF-β (p = 0.461) each decreased (Table 4 and Fig 1).

## Discussion

This study comprehensively compared standard immuno-hematological parameters among individuals with podoconiosis and healthy controls in Ethiopia. Podoconiosis patients showed

**Table 1. Sociodemographic Variables of Podoconiosis Patients and Controls in West Gojjam Zone, Ethiopia.**

| Variables | | Cases (N = 53) | Controls(N = 67) | | | | |
|---|---|---|---|---|---|---|---|
| Sex | N | % | N | % | X² | DF | P-value |
| Male | 34 | 64.2% | 45 | 67.2% | 0.12 | 1 | 0.730 |
| Female | 19 | 35.8% | 22 | 32.8% | | | |
| **Age (Years)** | | | | | | | |
| Mean (SD) | 49.94 (13.90) | | 41.46 (13.47) | | | | |
| 18–24 | 1 | 1.9% | 3 | 4.5% | 10.50 | 4 | **0.033** |
| 25–34 | 5 | 9.4% | 19 | 28.4% | | | |
| 35–44 | 16 | 30.2% | 23 | 34.3% | | | |
| 45–54 | 12 | 22.6% | 8 | 11.9% | | | |
| 55 and above | 19 | 57.6% | 14 | 20.9% | | | |
| **Education status** | | | | | | | |
| Illiterate | 45 | 84.9% | 43 | 64.2% | 6.50 | 1 | **0.011** |
| Literate | 8 | 15.1% | 24 | 35.8% | | | |
| **Occupation** | | | | | | | |
| Farmer | 53 | 100.0% | 57 | 85.1% | 8.63 | 4 | 0.071 |
| Student | 0 | 0.0% | 2 | 3.0% | | | |
| Merchant | 0 | 0.0% | 5 | 7.5% | | | |
| Others | 0 | 0.0% | 3 | 4.5% | | | |
| **Leg Washing Frequency** | | | | | | | |
| Two times/Day | 13 | 24.5% | 4 | 6.0% | 8.58 | 2 | **0.014** |
| Every Day | 32 | 60.4% | 48 | 71.6% | | | |
| > = 2 times/week | 8 | 15.1% | 15 | 22.4% | | | |
| **Shoe Wearing History** | | | | | | | |
| Yes | 47 | 88.7% | 66 | 98.5% | 5.20 | 1 | **0.023** |
| No | 6 | 11.3% | 1 | 1.5% | | | |
| **Age of Shoe Wearing** | | | | | | | |
| Mean (SD) | 27.10 (15.85) | | 27.06 (14.62) | | | | |
| 18–24 | 21 | 40.8% | 28 | 41.8% | 3.36 | 4 | 0.500 |
| 25–34 | 18 | 30.8% | 19 | 28.4% | | | |
| 35–44 | 8 | 15.8% | 11 | 16.4% | | | |
| 45–54 | 2 | 7.5% | 7 | 10.4% | | | |
| 55 and above | 4 | 5.0% | 2 | 3.0% | | | |
| **Type of Shoe Wearing** | | | | | | | |
| Protective shoe | 23 | 95.8 | 33 | 60 | 10.40 | 1 | **0.001** |
| Non-protective shoe | 1 | 4.2 | 22 | 40 | | | |
| **Marital Status** | | | | | | | |
| Married | 42 | 79.2% | 55 | 82.0% | 4.867 | 1 | 0.182 |
| Other | 11 | 20.8% | 12 | 18.0% | | | |
| **Floor Type** | | | | | | | |
| Covered | 0 | 0.0% | 3 | 4.5% | 2.43 | 1 | 0.119 |
| Uncovered | 53 | 100.0% | 64 | 95.5% | | | |
| **Religion** | | | | | | | |
| Orthodox | 53 | 100.0% | 66 | 98.5% | 0.80 | 1 | 0.372 |
| Others | 0 | 0.0% | 1 | 1.5% | | | |
| **Monthly Income** | | | | | | | |
| Mean (SD) | 208.09 (64.02) | | 266.01 (248.49) | | | | |
| <200 | 24 | 45.3% | 36 | 53.7% | 8.352 | 2 | 0.015 |

(*Continued*)

**Table 1.** (*Continued*)

| Variables | | Cases (N = 53) | | Controls(N = 67) | | | |
|---|---|---|---|---|---|---|---|
| 200–400 | 29 | 54.7% | 24 | 35.8% | | | |
| >400 | 0 | 0.0% | 7 | 10.4% | | | |
| **RDT (malaria)** | | | | | | | |
| Positive | ND | 0% | ND | 0% | | | |
| Negative | 53 | 100.00% | 67 | 100.00% | ND | 0 | ND |
| **Any Parasitic Infection** | | | | | | | |
| Any Parasite | 20 | 37.7% | 22 | 32.8% | 0.312 | 1.00 | 0.576 |
| No Parasite | 33 | 62.3% | 45 | 67.2% | | | |
| **Types of parasites identified** | | | | | | | |
| Hook worm | 15 | 75% | 7 | 31.8% | 14.29 | 7 | 0.046 |
| *G.lamblia* | 1 | 5% | 8 | 36.4% | | | |
| *E.histolytica* | 1 | 5% | 3 | 13.6% | | | |
| *E.vermicularis* | 1 | 5% | 0 | 0% | | | |
| *Taenia species* | 1 | 5% | 0 | 0% | | | |
| *S.stercoralis* | 1 | 5% | 1 | 4.5% | | | |
| *H.nana* | 0 | 0% | 1 | 4.5% | | | |
| *Mixed* | 0 | 0% | 2 | 9.1% | | | |
| **Blood group** | | | | | | | |
| A | 17 | 34.3% | 23 | 32.1% | 1.182 | 3.00 | 0.757 |
| AB | 6 | 7.5% | 5 | 11.3% | | | |
| B | 16 | 25.4% | 17 | 30.2% | | | |
| O | 14 | 32.8% | 22 | 26.4% | | | |

SD = Standard deviation, $X^2$ = chi-square, DF = degree of freedom, ND = Not detected

a significant decrease in average total WBC counts compared to controls. Moreover, a significant decrease in absolute neutrophil and eosinophil levels and a significant increase in absolute lymphocyte and monocyte levels were observed. Hemoglobin levels were significantly decreased among podoconiosis patients indicating borderline anemia according to WHO guidelines ($< 12$ g/dl for non-pregnant women and $<13$ g/dl for men) [21]. The mean hemoglobin value was 11.67 g/dl and 12.28 g/dl for non-pregnant women and men podoconiosis patients, respectively.

The final significant marker was the decreased percentage of CD4+ T cells among podoconiosis patients. While the cytokine profiles between patients and controls did not differ significantly, the trend toward lower TNF-α and IL-10 levels and higher TGF-β levels among patients may help postulate a biological mechanism for podocoinosis. In looking at the hematological profile for mild, moderate, and advanced stages of podoconiosis patients, RBCs, hemoglobin, and absolute lympocyte count decreased from mild to advanced while white blood cells, absolute monocytes, and absolute neutrophils increased from mild to advanced stages. The cytokine profile for podoconiosis patients by stage severity suggests trends of reduced IL-10 levels as the disease becomes more severe, while TNF-α appears to increase with severity. On the other hand, transforming growth factor-β (TGF-β) decreased in an early and advanced stages of the disease, similar to a study done in Ethiopia [14].

In uncovering potential explanations for the underlying mechanism behind the hematological data, many of the results had similarities to the profiles seen throughout various autoimmune disorders. Beginning with white blood cell count, it has been noted in a wide variety of

**Table 2. Hematological Parameters of Podoconiosis Patients and Controls in West Gojjam Zone, Ethiopia.**

| Variables | Patients (N = 53) | Controls (N = 67) | MD (95% CI) | p-value | AMD (95% CI) | p-value |
|---|---|---|---|---|---|---|
| | Mean (SD) | Mean (SD) | | | | |
| RBC (x10^12/L) | 4.85 (0.39) | 5.12 (0.50) | -0.27 (-0.44, -0.10) | 0.002 | -0.15 (-0.34, 0.03) * | 0.105 |
| WBC (x10^9/ L) | 5.86 (2.00) | 7.39 (2.63) | -1.53 (-2.39, -0.66) | 0.001 | -1.15 (-2.09, -0.21) ** | 0.017 |
| HGB (g/dL) | 12.06 (0.94) | 14.14 (1.59) | -2.07 (-2.57, -1.59) | <0.001 | -1.73 (-2.28, -1.19) *** | <0.001 |
| PLT (x10^9/ L) | 248.66 (97.62) | 261.83 (91.20) | -13.17 (-47.42, 21.07) | 0.448 | -25.45 (-60.68, 9.79) **** | 0.257 |
| MPV (fl) | 13.44 (13.90) | 11.34 (1.61) | 2.10 (-1.42, 5.63) | 0.239 | 1.78 (-1.98, 5.53) ***** | 0.35 |
| Abs Lymphocyte (x10^9/ L) | 2.99 (0.92) | 2.09 (1.13) | 0.90 (0.45, 1.35) | <0.001 | 0.98 (0.50, 1.46) ****** | <0.001 |
| Abs Monocyte (x10^9/ L) | 1.26 (0.86) | 0.86 (0.92) | 0.40 (0.07, 0.72) | 0.016 | 0.54 (0.22, 0.85) ******* | 0.001 |
| Abs Neutrophil (x10^9/ L) | 0.72 (0.85) | 4.38 (2.33) | -3.67 (-4.34, -2.99) | <0.001 | -3.42 (-4.15, -2.69) ******** | <0.001 |
| Abs Eosinophil (x10^9/ L) | 0.33 (0.37) | 0.51 (0.48) | -0.18 (-0.34, -0.03) | 0.026 | -0.20 (-0.37, -0.03) ********* | 0.019 |
| Abs Basophil (x10^9/ L) | 0.023 (0.037) | 0.025 (0.058) | -0.001 (-.02, 0.02) | 0.871 | -0.004 (-0.02, 0.02) ********** | 0.691 |
| MCV (fl) | 93.98 (3.47) | 92.18 (4.32) | 1.79 (0.35, 3.24) | 0.015 | 1.88 (0.38, 3.81) ********** | 0.014 |
| MCH (pg) | 24.85 (0.97) | 27.66 (1.85) | -2.81 (-3.37, -2.25) | <0.001 | -2.81 (-3.41, -2.11) ********** | <0.001 |
| MCHC (g/dl) | 26.45 (0.38) | 30.03 (2.03) | -3.58 (-4.14, -3.02) | <0.001 | -3.57 (-4.19, -2.94) ************ | <0.001 |
| CD4 # | 673.58 (224.46) | 681.13 (246.78) | -7.55 (-93.89, 78.80) | 0.863 | 40.06 (-54.94, 135.06) ************* | 0.405 |
| CD4% | 34.89 (7.54) | 37.61 (8.56) | -2.72 (-5.68, 0.24) | 0.071 | -3.39 (-6.39, -0.40) ************** | 0.027 |

RBC = Red Blood Cell Count, WBC = White Blood Cell Count, HGB = Hemoglobin, PLT = Platelet Count, MPV = Mean Platelet Volume, Abs = Absolute

MCV = Mean Corpuscular Volume, MCH = Mean Corpuscular Hemoglobin, MCHC = Mean Corpuscular Hemoglobin Concentration, AMD = Adjusted mean difference

All AMD were adjusted for between-subject effects with p < 0.5

*Adjusted for Age, Education Level, Leg Washing, Shoe Wearing, and Monthly Income

**Adjusted for Education Level, Leg Washing, Shoe Wearing, Monthly income, Occupation, and Marital Status

***Adjusted for Age, Leg Washing, Shoe Wearing, Monthly Income, and Marital Status

****Adjusted for Shoe Wearing and Education Level

*****Adjusted for Age and Shoe Wearing

******Adjusted for Shoe Wearing, Education Level, Occupation, Marital Status, and Monthly Income

*******Adjusted for Shoe Wearing and Monthly Income

********Adjusted for Age, Occupation, Marital Status, Leg Washing, and Monthly Income

*********Adjusted for Occupation, Monthly Income, Marital Status, and Shoe Wearing

**********Adjusted for Education Level, Monthly Income, Marital Status, and Shoe Wearing

***********Adjusted for Monthly Income, Marital Status, Education Level, and Leg Washing

************Adjusted for Age, Occupation, Education Level, Marital Status, and Monthly Income

*************Adjusted for Age, Occupation, Leg Washing, and Monthly Income

**************Adjusted for Age, Occupation, Leg Washing, Shoe Wearing, and Marital Status

*************** Adjusted for Shoe Wearing and Educational Level

autoimmune disorders to have a low white blood cell count [26] as a frequent symptom, such as in systemic lupus erythematosus (SLE) as one particular study had a prevalence of 50% of patients with leukopenia [27]. It is suggested that the low white blood cell count in autoimmune conditions can be explained by the body attacking its own healthy white blood cells [27]. In a further breakdown of patients with SLE, neutropenia was found in 20% of cases [27] while an additional study discovered that neutropenia is the principal cytopenia for patients with rheumatoid arthritis (RA) [28]. One proposed explanation is that within autoimmune disorders such as RA and SLE, antineutrophil cytoplasmic antibodies are produced leading to neutrophil apoptosis [28]. We observed a significant reduction in mean absolute neutrophil counts in patients with podoconiosis compared to the control group. This decrease may be linked to the presence of autoimmune neutropenia (AIN) in these patients, although we did

**Table 3. Immunological Factors of Podoconiosis Patients and Controls in West Gojjam Zone, Ethiopia.**

|  | Patients (N = 53) | Controls (N = 67) | MD (95% CI) | p-value | AMD (95% CI) | p-value |
|---|---|---|---|---|---|---|
| Variables | Mean (SD) | Mean (SD) |  |  |  |  |
| ln [TNFa] | 0.47 (1.14) | 0.78 (1.17) | -0.31 (-0.73, 0.11) | 0.147 | -0.26 (-0.71, 0.18) * | 0.246 |
| ln [IL17] | 0.20 (0.73) | 0.23 (0.58) | -0.02 (-0.26,0.21) | 0.842 | 0.02, (-0.23, 0.27) ** | 0.885 |
| ln [IL10] | 1.33 (2.14) | 1.54 (2.07) | -0.21 (-0.97, 0.56) | 0.592 | -0.31 (-1.14, 0.52) *** | 0.465 |
| ln [IL4] | 0.69 (1.45) | 0.61 (1.41) | 0.09 (-0.43, 0.61) | 0.739 | 0.03 (-0.53, 0.59) **** | 0.903 |
| ln [IFN-y] | 0.16 (0.85) | 0.15 (0.87) | 0.01 (-0.29, 0.33) | 0.929 | 0.05 (-0.25, 0.34) ***** | 0.749 |
| ln [IL6] | 0.90 (0.82) | 0.81 (0.87) | 0.09 (-0.22, 0.40) | 0.552 | -0.01 (-0.36, 0.34) ****** | 0.972 |
| ln [TGF-B] | 6.73 (0.32) | 6.66 (0.35) | 0.07 (-0.05, 0.19) | 0.279 | 0.07 (-0.06, 0.21) ******* | 0.292 |

All Immunological Variables were logarithmically adjusted using ln[x], TNF-a = Tumor Necrosis Factor alpha, IL = Interleukin, IFN-y = Interferon gamma,

TGF-B = Transforming Growth Factor Beta, AMD = Adjusted Mean difference

* Adjusted for Age, Education Level, Occupation, Marital Status, and Monthly Income

**Adjusted for Age, Education Level, Occupation, Monthly Income, and Shoe Wearing

***Adjusted for Occupation, Shoe Wearing, and Leg Washing

****Adjusted for Let Washing, Marital Status, and Monthly Income

*****Adjusted for Marital Status, Shoe Wearing, and Monthly Income

******Adjusted for Age, Marital Status, Occupation, Leg Washing, and Monthly Income

*******Adjusted for Occupation, Education Level, Leg Washing, and Shoe Wearing

not examine any autoimmune variables. Individuals with AIN typically have neutrophil counts lower than $1.5 \times 10^9/L$, a condition associated with the peripheral destruction of neutrophils due to autoantibodies targeting the patient's own neutrophils [29,30]. A genomic study on podoconiosis revealed that the patients had higher RNA transcript levels for activation, scavenger receptor and apoptosis markers while lower levels of markers for histones, T cell receptors, as well as variable and constant immunoglobulins compared to healthy controls. It seems plausible to suggest that alongside these other autoimmune diseases, podoconiosis could have a similar mechanism that could explain why patients had neutrophil scarcity. The association of

**Table 4. Hematological Profile of Podoconiosis Patients by Severity in West Gojjam Zone, Ethiopia.**

|  | Mild (n = 5) | Moderate (n = 21) | Advanced (n-27) | F | DF | p-value |
|---|---|---|---|---|---|---|
| Variables | Mean (SD) | Mean (SD) | Mean (SD) |  |  |  |
| RBC (x10^12/L) | 5.19 (0.33) | 4.75 (0.38) | 4.86 (0.38) | 4.82 | 3 | **0.003** |
| WBC (x10^9/L) | 4.24 (0.76) | 6.40 (2.07) | 5.72 (1.98) | 5.23 | 3 | **0.002** |
| HGB (g/dL) | 12.98 (0.87) | 11.81 (0.84) | 12.08 (0.95) | 24.77 | 3 | **<0.001** |
| PLT (x10^9/L) | 259.20 (51.64) | 233.62 (95.29) | 258.41 (106.59) | 0.49 | 3 | 0.693 |
| MPV (fl) | 10.96 (0.25) | 12.11 (2.15) | 14.93 (19.44) | 0.93 | 3 | 0.427 |
| Abs Lymphocyte (x10^9/L) | 2.85 (0.72) | 3.26 (1.29) | 1.14 (0.85) | 5.85 | 3 | **0.001** |
| Abs Monocyte (x10^9/L) | 0.74 (0.62) | 1.54 (0.85) | 1.14 (0.85) | 3.47 | 3 | **0.018** |
| Abs Neutrophil (x10^9/L) | 0.72 (0.54) | 0.62 (0.69) | 0.79 (1.01) | 38.82 | 3 | **<0.001** |
| Abs Eosinophil (x10^9/L) | 0.26 (0.15) | 0.40 (0.52) | 0.28 (0.24) | 2.04 | 3 | 0.112 |
| Abs Basophil (x10^9/L) | 0.02 (0.03) | 0.02 (0.03) | 0.03 (0.04) | 0.08 | 3 | 0.971 |
| MCV (fl) | 93.46 (2.29) | 94.04 (3.32) | 94.03 (3.85) | 2.03 | 3 | 0.114 |
| MCH (pg) | 25.00 (0.91) | 24.86 (0.93) | 24.81 (1.04) | 32.79 | 3 | **<0.001** |
| MCHC (g/dl) | 26.74 (0.51) | 26.47 (0.38) | 26.38 (0.34) | 52.60 | 3 | **<0.001** |
| CD4 # | 514.60 (78.16) | 702.14 (219.13) | 680.81 (239.40) | 0.87 | 3 | 0.457 |
| CD4% | 35.07 (4.88) | 33.84 (8.03) | 35.66 (7.68) | 1.29 | 3 | 0.281 |

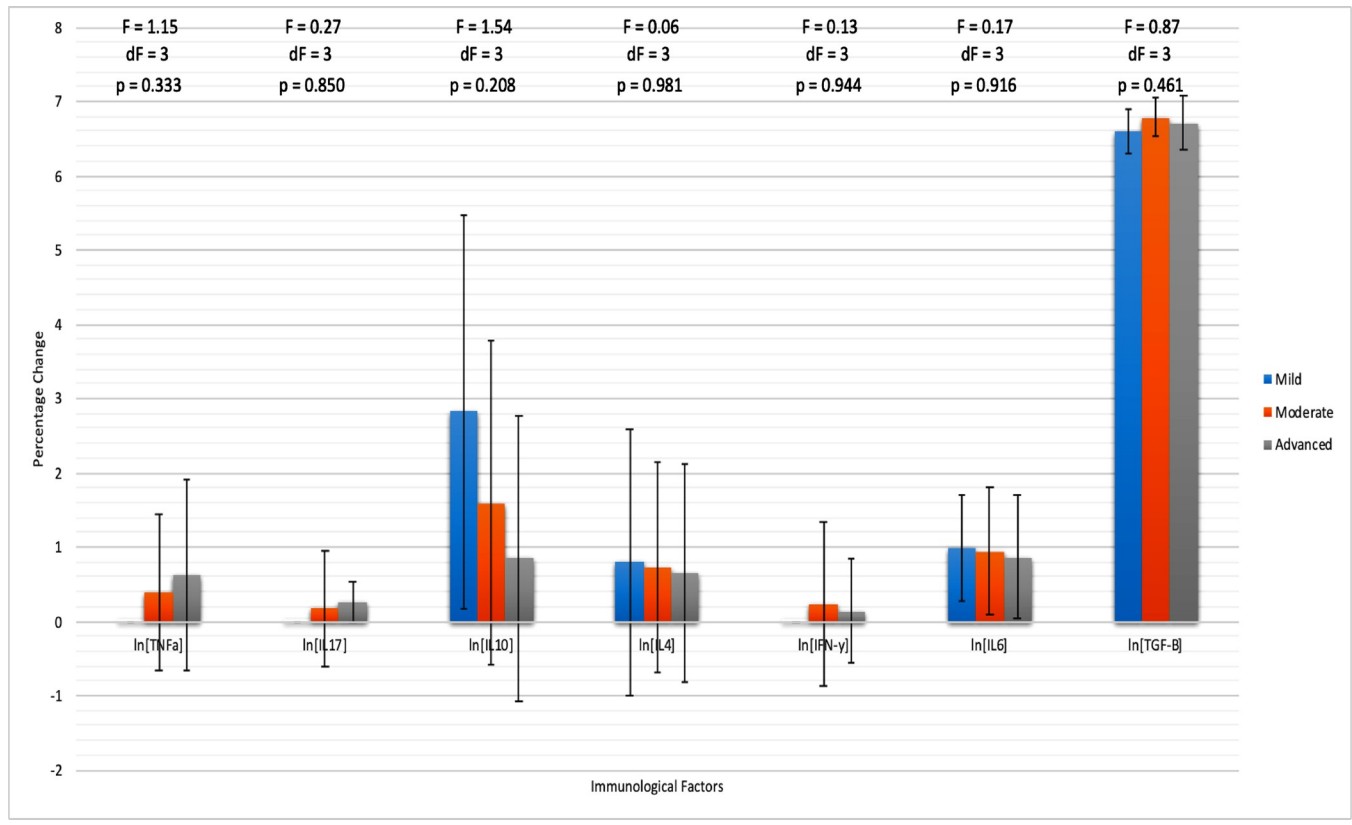

**Fig 1. Cytokine Profile of Podoconiosis Patient by Stage suggests IL-4, IL-10, and IL-6 reduce with severity while TNF-α and IL-17 increase with severity.** Immunological profile data for Podoconiosis patients at different stages with mild (n = 5), moderate (n = 21), and advanced (n = 27). ANOVA analysis occurred, with F value, degree of freedom, and p-value recorded. Each immunological factor has been adjusted to a logarithmic scale using ln(x) to give the percentage change as the dependent variable.

silica exposure with autoimmunity, which is an agreed etiology for podoconiosis, is demonstrated in a mice model favoring our argument that podoconiosis could be an autoimmune condition [31].

An explanation for the reduced eosinophil count is not fully understood, but previous studies suggested that eosinophils, which are effector cells frequently seen in tissue remodeling, can have damaging effects during the inflammatory process and reduced in the peripheral blood [32,33]. Additionally, eosinophils and lymphocytes have been noted to have an important role in initiating and maintaining unexpected inflammatory responses in autoimmune diseases like SLE, inflammatory bowel disease (IBD), RA, and even asthma [34]. These may indicate that the chronic inflammation seen in podoconiosis patients has either a normal or amplified inflammation response by lymphocytes and monocytes. Monocytes have a similarly important role in inflammatory response in producing macrophages, so their increased levels likely indicate an important role within the disease pathogenesis [35].

On the other hand, the most likely causes of the observed low hemoglobin in podoconiosis patients might be related to the release of inflammatory proteins like hepcidin and lactoferrins which have high affinity to iron than transferrin and result in low serum iron and inhibit iron absorption [36,37]. The other possible causes might be related to the activation of pro-inflammatory cytokines suppress erythropoietin response to anemia and shorten RBC half-life [36]. Low hemoglobin levels indicate hemolysis caused by chronic inflammation, as seen with a significant decrease in from mild to advanced stage [20]. Further investigation is warranted to

rule out or rule in autoimmune destruction of erythrocytes. Low hemoglobin could alternatively result from malnutrition caused by the severe stigmatization and limited mobility of podoconiosis patients [20]. Finally, the reduced CD4% can be potentially explained by the observation of a mutation within the HLA II gene cluster negatively affecting the regulatory role of CD4+ T helper cells [38], which could be additional insight into the role of chronic immune activation affecting the CD4 cells [39].

While the evidence for the hematological profile is more thoroughly backed by the data, the trends in the cytokine profiles among individuals with podoconiosis and healthy controls can supplement the potentially autoimmune-related mechanism. TNF-α, for instance, has been seen to increase in response to reactive oxygen species (ROS) in studies focusing on silica-induced inhalation toxicity [40]. Even though podoconiosis patients compared to controls showed a reduction in TNF-α levels, when looking at the data by disease stage, the TNF-α levels increase in relation to the severity. This could be explained by the increasing amount of ROS that accumulates through the chronic inflammation and tissue destruction process. TNF-α is additionally a key inflammatory cytokine associated with RA and SLE among other autoimmune diseases with low TNF-α playing a role in susceptibility as well as in the clinical manifestation of the diseases [41].

Although many of the other factors match the typical autoimmune profile, IL-10 which is typically associated with tissue repair does not follow this pattern. The literature has shown that SLE patients have elevated levels of IL-10 [42]. However, with inflammatory markers down, it makes sense that there are lower than normal IL-10 levels as it is typically seen as a potent anti-inflammatory and could explain certain differences in presentation between SLE and podoconiosis [41]. As for TGF-β levels, while not significant, our data suggests that TGF-β is higher among podoconiosis patients than controls which opposes the previous finding measuring reduced TGF-β among other oxidative stress markers [14] whereas increased levels of TGF-β and TNF-α, have been seen in silica exposure induced autoimmunity in a mice model [31]. A recent study found a link between the ABO blood group and cytokine levels in COVID-19 patient outcomes. The researchers reported that individuals with blood group O had higher cytokine levels, which were associated with better outcomes compared to those with blood groups A, B, or AB [43]. We also included the ABO blood group in our model and adjusted our main outcome variables accordingly.

## Limitations

The findings of this study should be interpreted with caution. First, as a cross-sectional study, it limits our ability to draw causal relationships between immunohematology parameters and podoconiosis. Our smaller sample size prevents us from seeing significant differences in most cytokine profiles, and a future high-powered study is recommended. Changes in the immunohematology profile can be influenced by various diseases other than podoconiosis. While it's challenging to rule out other factors, we examined common intestinal parasites and malaria infections as indicators of infection. These were used to adjust our outcome variable. There are further limitations in measuring serum cytokine levels individually, as this approach comes with inherent challenges. Cytokines are pleiotropic by nature, meaning they can have diverse effects, and their expression patterns can change dynamically, regardless of any illness [44,45]. For instance, variations in hormone levels, such as cortisol, have been shown to inversely affect the expression of several inflammatory cytokines, including IFNγ, TNFα, IL-1, and IL-12 [45]. We did not assess blood hormone levels, which could be crucial for accurately interpreting inflammatory markers. Additionally, there is no well-established reference range for categorizing normal and abnormal cytokine levels.

## Conclusion

Our findings showed significant changes in hematological parameters, particularly a decrease in absolute neutrophil counts among individuals with podoconiosis, suggesting a possible link to an autoimmune process. Additionally, we observed lower hemoglobin levels in podoconiosis patients compared to healthy controls. However, the cytokine profiles did not show any significant differences between the podoconiosis group and the healthy controls. Future mechanistic research is necessary to confirm our findings and clarify the role of reduced neutrophils in the development of podoconiosis.

## Acknowledgments

We thank Addis Ababa University, College of Health Sciences, Department of Medical Laboratory Sciences and Amhara Public Health Institute for allowing us to conduct the research. We also thank Ethiopian Blood and Tissue Bank service for their collaboration to use the facility for ELISA testing. Our deepest gratitude goes to IOCC Ethiopia for helping us in the case identification. We would like to acknowledge data collectors for accurate data collection, study participants for voluntary participation and provision of their consent to be interviewed and provide biological samples. The PI thanks Debre Markos University for providing the scholarship opportunity to join MSc in Clinical Laboratory Science program at Addis Ababa University.

## Author Contributions

**Conceptualization:** Aytenew Atnaf, Kassu Desta, Bineyam Taye, Aster Tsegaye.

**Data curation:** Aytenew Atnaf, Ryan Oakes, Kassu Desta, Bineyam Taye, Aster Tsegaye.

**Formal analysis:** Aytenew Atnaf, Ryan Oakes, Bruno T. Scodari, Kassu Desta, Bineyam Taye, Aster Tsegaye.

**Funding acquisition:** Aytenew Atnaf, Kassu Desta, Bineyam Taye, Aster Tsegaye.

**Investigation:** Aytenew Atnaf.

**Methodology:** Aytenew Atnaf, Kassu Desta, Bineyam Taye, Aster Tsegaye.

**Project administration:** Aytenew Atnaf, Kassu Desta, Bineyam Taye, Aster Tsegaye.

**Resources:** Aytenew Atnaf, Kassu Desta, Bineyam Taye, Aster Tsegaye.

**Software:** Bruno T. Scodari.

**Supervision:** Kassu Desta, Bineyam Taye, Aster Tsegaye.

**Validation:** Aytenew Atnaf, Bruno T. Scodari, Kassu Desta, Bineyam Taye, Aster Tsegaye.

**Visualization:** Aytenew Atnaf, Bruno T. Scodari, Kassu Desta, Bineyam Taye, Aster Tsegaye.

**Writing – original draft:** Aytenew Atnaf, Ryan Oakes, Bruno T. Scodari, Kassu Desta, Bineyam Taye, Aster Tsegaye.

**Writing – review & editing:** Aytenew Atnaf, Ryan Oakes, Bruno T. Scodari, Kassu Desta, Bineyam Taye, Aster Tsegaye.

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
