## [Decision Letter · Decision Letter 0]

5 Aug 2024

Dear Mr Atnaf,

Thank you very much for submitting your manuscript "Hematological and Immunological Evidence for Podoconiosis Autoimmune Disease Pathogenesis, West Gojjam Zone, Ethiopia: A Comparative Cross-Sectional Study" for consideration at PLOS Neglected Tropical Diseases. As with all papers reviewed by the journal, your manuscript was reviewed by members of the editorial board and by several independent reviewers. In light of the reviews (below this email), we would like to invite the resubmission of a significantly-revised version that takes into account the reviewers' comments.

We cannot make any decision about publication until we have seen the revised manuscript and your response to the reviewers' comments. Your revised manuscript is also likely to be sent to reviewers for further evaluation.

Sincerely,

Marcelo Larami Santoro, DVM, MSc, PhD

Academic Editor

Justin Remais

Section Editor

Reviewer's Responses to Questions

**Key Review Criteria Required for Acceptance?**

**Methods**

-Are the objectives of the study clearly articulated with a clear testable hypothesis stated?

-Is the study design appropriate to address the stated objectives?

-Is the population clearly described and appropriate for the hypothesis being tested?

-Is the sample size sufficient to ensure adequate power to address the hypothesis being tested?

-Were correct statistical analysis used to support conclusions?

-Are there concerns about ethical or regulatory requirements being met?

Reviewer #1: The objective of the article clearly articulated.

The study design has a detailed procedure on the laboratory evaluation but the authors failed to mention on how the diagnosis and clinical staging is done, how the sampling is calculated (What was the logic behind recruiting recruiting 53 podoconiosis patients and 67 controls), how is podoconiosis diagnosed and who has diagnosed them. Additionally how how is the staging done (mild moderate and severe) and the criteria used is not clear. Taking the multiple variables comparison to consideration the very small sample size of patients (5 mild, 21 moderate, 27 severe) is clearly one major critical limitation of the study. The small sample size and the variation of the patient staging makes it insufficient to ensure adequate power for the hypothesis being tested. 

The author stated the ethical approval for the study is secured from the regional health bureau (which as far as my knowledge goes the bureau does not have a committee....they only provide support letter), while the authers affiliation is in the university (and it is known both Debremarkos University and AAU have IRB process (mandated by national ethical committee). Most importantly the author claims the data is collected in the IOCC podoconiosis treatment centers but no one from the NGO is mentioned as co-author

Reviewer #2: (No Response)

Reviewer #3: The article is generally interesting, as there is still much to be learned about podoconiosis. The objectives are clear and the research is well developed, although, as the authors state at the end of the manuscript, proper footwear and education are the pillars of prevention and treatment, so in the future it may be more useful to address the next steps in this direction. 

The study design is adequate to address the stated objectives and the population is clearly described for the hypothesis. However, there are still data that could be more detailed. As the authors explain in the discussion, "low hemoglobin levels reflects the hemolysis that occurs through chronic inflammation, as additionally seen with RBCs significantly decreasing by stage from mild to advanced". In this regard, for the study of anemia, if present, LDH and total bilirubin should have been studied as the presence of hemolytic anemia is of concern. 

The sample size should be increased in future investigations, but enough to ensure adequate power to address the hypothesis being tested.

The statistical analysis is correct and there are no ethical or regulatory concerns.

**Results**

-Does the analysis presented match the analysis plan?

-Are the results clearly and completely presented?

-Are the figures (Tables, Images) of sufficient quality for clarity?

Reviewer #1: The result is presented according to the analysis plan and result is completely presented.

Reviewer #2: (No Response)

Reviewer #3: The analysis match the analysis plan. Results are clearly and completely presented and figures are of sufficient quality for clarity.

Nonetheless, regarding age, cases were aproximately 8 years older than controls (49 vs 41yo). As far as we know, direct contact for a longer period (age) increases the risk for podoconiosis. Also age could increase the risk for other diseases and anemia due to other causes such as cancer. Authors should specify comorbidity or concomitant illnesses of the studied population. Anemia could also be explained by the presence of parasites (hookworm and other soil-transmitted helminths). In the table, it is described the presence of "any parasite" in 37,7% of cases and 32,8% of controls. Even if no statistically significant, authors should describe what parasites they found in stools or other tests.

Additionally, it is remarkable that 24% of cases (13p) declared washing their legs twice a day vs. 6% (4p) of the controls. How do the authors explain this unexpected finding?

**Conclusions**

-Are the conclusions supported by the data presented?

-Are the limitations of analysis clearly described?

-Do the authors discuss how these data can be helpful to advance our understanding of the topic under study?

-Is public health relevance addressed?

Reviewer #1: In general the conclusion is not supported by the finding and the author bold statement mentioned starting on the title are based on false premiss. The limitation of the study such as he small sample size, unclear diagnostic and staging criteria and the fact all the authors are laboratory personals with no clinical background for participant recruitment is not mentioned.

Reviewer #2: (No Response)

Reviewer #3: Conclusions are supported by the data presented and limitations are clearly described.

**Editorial and Data Presentation Modifications?**

Reviewer #1: The title is very bold statement to attract the attention of experts who are working on the disease but when I dive deep to findings it does not have the evidence to support the title. 

The authors look does not have much experience/knowledge on podoconiosis which is apparent to see in the introduction wrong conclusions like ‘ Podoconiosis classified as a Neglected Tropical Disease 45 (NTD) by the World Health Organization in 2011’ which is not right statement …..podo is not added under the list of WHO NTD list. And conclusive statements like “Podoconiosis, ….is a geochemical disease caused by alkali volcanic-originating soil’ ...this is so far a hypothesis (the cause effect relation of podoconiosis and soil is not confirmed) needs additional study to conclude.

Reviewer #2: (No Response)

Reviewer #3: (No Response)

**Summary and General Comments**

Reviewer #1: Review: 'Hematological and Immunological Evidence for Podoconiosis Autoimmune Disease Pathogenesis'

Title and Introduction

The title "Hematological and Immunological Evidence for Podoconiosis Autoimmune Disease Pathogenesis" makes a bold claim that is not substantiated by the findings within the study. This creates a misleading impression for experts interested in the disease. The introduction contains inaccuracies, such as the incorrect statement that "Podoconiosis was classified as a Neglected Tropical Disease (NTD) by the World Health Organization in 2011." In reality, podoconiosis has not been added to the WHO NTD list. Additionally, the statement “Podoconiosis...is a geochemical disease caused by alkali volcanic-originating soil” is presented as a definitive conclusion, whereas this remains a hypothesis that requires further research to confirm.

Study Design and Methodology

The study includes a detailed laboratory procedure but lacks critical information on the diagnosis and clinical staging of podoconiosis, as well as the rationale behind the sampling size. Key methodological details, such as the criteria for diagnosing podoconiosis, who performed the diagnoses, and how staging (mild, moderate, severe) was determined, are missing. The small sample size (53 podoconiosis patients and 67 controls) and the distribution across different stages of the disease (5 mild, 21 moderate, 27 severe) are significant limitations. These limitations undermine the study’s ability to ensure adequate power for the hypothesis being tested.

Ethical Considerations

The study claims to have received ethical approval from the regional health bureau, which does not typically have a dedicated ethical review committee. Given that the authors are affiliated with a university, it is expected that approval would be sought from the institutional review board (IRB) at either Debremarkos University or Addis Ababa University, both of which have established IRB processes mandated by the national ethical committee.

Study Findings and Conclusion

The conclusions of the study are not supported by the findings. The limitations of the study, such as the small sample size, unclear diagnostic and staging criteria, and the lack of clinical expertise among the authors, are not adequately addressed. The discussion attempting to link podoconiosis to autoimmune disease lacks scientific grounding. Specific issues include:

1. Hematologic Profiles: The study indicates that podoconiosis patients have decreased hematologic profiles compared to controls, but these values remain within normal ranges, suggesting that these patients are not pancytopenic. A reduced hematologic profile can result from various conditions such as malnutrition, drugs, infections, or malignancies, not exclusively autoimmune diseases.

2. Other Causes: The study does not rule out other potential causes for the decreased hematologic profiles in podoconiosis patients, such as hookworm-induced anemia due to barefoot exposure, which the study acknowledges as common among podoconiosis patients.

3. Contradictory Findings: The report that absolute lymphocyte and monocyte counts are significantly higher in podoconiosis patients contradicts the notion of pancytopenia and does not support the autoimmune disease hypothesis.

4. Immunologic Profiles: The observed reduction in TNF-α among podoconiosis patients contradicts the argument for an autoimmune disease. The decrease in CD4 count, proposed as evidence for autoimmune disease, overlooks the fact that in humans, CD4 depletion (as seen in HIV infection) usually ameliorates clinical activity rather than exacerbating it.

5. Silica Association: The claim of an association between podoconiosis and silica is not supported by recent geochemical studies, which show no significant correlation. The mean silica particle concentration is slightly higher in non-endemic areas for podoconiosis, refuting the hypothesis of silica involvement. (actually the mean silica particle concentration is slightly higher in non podoconiosis endemic area) (Cooper, Jamey N, and Kevin E Nick. “A geochemical and mineralogical characterization of soils associated with podoconiosis.” Environmental geochemistry and health vol. 45,11 (2023): 7791-7812.)

Summary

The title of the study is misleading, and the conclusions drawn do not align with the actual findings. The study's significant methodological flaws and unsupported claims undermine its contribution to the understanding of podoconiosis. The authors need to address these limitations and provide a more balanced and accurate interpretation of their data.

Reviewer #2: (No Response)

Reviewer #3: (No Response)

PLOS authors have the option to publish the peer review history of their article (what does this mean?). If published, this will include your full peer review and any attached files.

Reviewer #1: Yes: Wendemagegn Enbiale

Reviewer #2: Yes: Mwesigye Vicent

Reviewer #3: No
---

## [Decision Letter · Decision Letter 1]

22 Oct 2024

PNTD-D-24-00819R1Hematological and Immunological Evidence for Podoconiosis Autoimmune Disease Pathogenesis, West Gojjam Zone, Ethiopia: A Comparative Cross-Sectional StudyPLOS Neglected Tropical Diseases Dear Dr. Atnaf, Thank you for submitting your manuscript to PLOS Neglected Tropical Diseases. After careful consideration, we feel that it has merit but does not fully meet PLOS Neglected Tropical Diseases's publication criteria as it currently stands. Therefore, we invite you to submit a revised version of the manuscript that addresses the points raised during the review process. Please submit your revised manuscript within 30 days Nov 21 2024 11:59PM. If you will need more time than this to complete your revisions, please reply to this message or contact the journal office at plosntds@plos.org. Please include the following items when submitting your revised manuscript:*
A rebuttal letter that responds to each point raised by the editor and reviewer(s). You should upload this letter as a separate file labeled 'Response to Reviewers'. This file does not need to include responses to any formatting updates and technical items listed in the 'Journal Requirements' section below.*
A marked-up copy of your manuscript that highlights changes made to the original version. You should upload this as a separate file labeled 'Revised Manuscript with Track Changes'.*
An unmarked version of your revised paper without tracked changes. You should upload this as a separate file labeled 'Manuscript'. If you would like to make changes to your financial disclosure, competing interests statement, or data availability statement, please make these updates within the submission form at the time of resubmission. Guidelines for resubmitting your figure files are available below the reviewer comments at the end of this letter. We look forward to receiving your revised manuscript. Kind regards, Marcelo Larami Santoro, DVM, MSc, PhDAcademic EditorPLOS Neglected Tropical Diseases Justin RemaisSection EditorPLOS Neglected Tropical Diseases

Shaden Kamhawi

co-Editor-in-Chief

Paul Brindley

co-Editor-in-Chief

 **Journal Requirements:** **Additional Editor Comments (if provided):****Reviewers' comments:** Reviewer's Responses to Questions

**Key Review Criteria Required for Acceptance?**

**Methods**

-Are the objectives of the study clearly articulated with a clear testable hypothesis stated?

-Is the study design appropriate to address the stated objectives?

-Is the population clearly described and appropriate for the hypothesis being tested?

-Is the sample size sufficient to ensure adequate power to address the hypothesis being tested?

-Were correct statistical analysis used to support conclusions?

-Are there concerns about ethical or regulatory requirements being met?

Reviewer #1: comments from the first review are addressed

Reviewer #2: (No Response)

Reviewer #3: (No Response)

**Results**

-Does the analysis presented match the analysis plan?

-Are the results clearly and completely presented?

-Are the figures (Tables, Images) of sufficient quality for clarity?

Reviewer #1: comments from the first review are addressed

Reviewer #2: (No Response)

Reviewer #3: The paper has been clarified in general. However, there is still missing some relevant information.

The authors have decided not to include details about parasitological examinations because they want to present them in a separate publication. However, results seem to be incomplete or ambiguous in this case.

Regarding leg-washing practices, the authors state that " our data showed that podoconiosis patients wash their legs more

frequently than controls. This increased frequency is likely due to the need to alleviate lymphedema-related

discomfort. Frequent washing helps reduce pain and is also practical for patients who are less involved in

physically demanding activities, such as farming, due to their condition."

This apparently is a surmise or hypothesis. Authors should consider whether religion might play a role. Muslim population practice ablutions but it doesn't imply that it is a thorough wash that may remove direct contact of soil with the skin.

**Conclusions**

-Are the conclusions supported by the data presented?

-Are the limitations of analysis clearly described?

-Do the authors discuss how these data can be helpful to advance our understanding of the topic under study?

-Is public health relevance addressed?

Reviewer #1: comments from the first review are addressed

Reviewer #2: (No Response)

Reviewer #3: (No Response)

**Editorial and Data Presentation Modifications?**

Reviewer #1: (No Response)

Reviewer #2: (No Response)

Reviewer #3: (No Response)

**Summary and General Comments**

Reviewer #1: most of the comments from the first review are addressed

Please again correct the misinformation which states Podoconiosis is 'classified as a Neglected Tropical

64 Disease (NTD) by the World Health Organization in 2011' ....

Reviewer #2: (No Response)

Reviewer #3: (No Response)

PLOS authors have the option to publish the peer review history of their article (what does this mean?). If published, this will include your full peer review and any attached files.

Reviewer #1: **Yes: **Wendemagegn Enbiale

Reviewer #2: **Yes: **MWESIGYE VICENT

Reviewer #3: No

---

## [Decision Letter · Decision Letter 2]

26 Nov 2024

Dear Mr Atnaf,

We are pleased to inform you that your manuscript 'Hematological and Immunological Profiles of Podoconiosis Patients in West Gojjam Zone, Ethiopia: A Comparative Cross-Sectional Study' has been provisionally accepted for publication in PLOS Neglected Tropical Diseases.

Best regards,

Marcelo Larami Santoro, DVM, MSc, PhD

Academic Editor

Justin Remais

Section Editor

Shaden Kamhawi

co-Editor-in-Chief

Paul Brindley

co-Editor-in-Chief

The manuscript may be accepted for publication. However, one of the reviewers noted that the authors did not specify the types of parasites identified in the stool samples, as their presence could influence the analysis of hematological data. Therefore, to proceed with publication, the authors are requested to include in Table 1 the species of parasites identified in the stool samples and their respective frequencies, similarly as the distribution of blood group types.

Reviewer's Responses to Questions

**Key Review Criteria Required for Acceptance?**

**Methods**

-Are the objectives of the study clearly articulated with a clear testable hypothesis stated?

-Is the study design appropriate to address the stated objectives?

-Is the population clearly described and appropriate for the hypothesis being tested?

-Is the sample size sufficient to ensure adequate power to address the hypothesis being tested?

-Were correct statistical analysis used to support conclusions?

-Are there concerns about ethical or regulatory requirements being met?

Reviewer #2: (No Response)

Reviewer #3: (No Response)

Reviewer #4: -

**Results**

-Does the analysis presented match the analysis plan?

-Are the results clearly and completely presented?

-Are the figures (Tables, Images) of sufficient quality for clarity?

Reviewer #2: (No Response)

Reviewer #3: The authors have added scarce information about parasitological examinations, as they say, in page 14, lines 246-247). However, since the presence of parasites is higher in the podoconosis group, I consider it would be relevant to detail which parasites are involved, even if the sample is small, specially when the ultimate aim of the study is to assess the haematological profile. Hookworm infestation is a mayor cause of anemia in the rural areas.

Reviewer #4: -

**Conclusions**

-Are the conclusions supported by the data presented?

-Are the limitations of analysis clearly described?

-Do the authors discuss how these data can be helpful to advance our understanding of the topic under study?

-Is public health relevance addressed?

Reviewer #2: (No Response)

Reviewer #3: (No Response)

Reviewer #4: -

**Editorial and Data Presentation Modifications?**

Reviewer #2: (No Response)

Reviewer #3: (No Response)

Reviewer #4: Requested modifications have been carried out by the authors.

**Summary and General Comments**

Reviewer #2: (No Response)

Reviewer #3: (No Response)

Reviewer #4: The text can be published.

PLOS authors have the option to publish the peer review history of their article (what does this mean?). If published, this will include your full peer review and any attached files.

Reviewer #2: **Yes: **VICENT MWESIGYE

Reviewer #3: No

Reviewer #4: No

---

## [Editor Report · Acceptance letter]

10 Dec 2024

Dear Mr Atnaf,

We are delighted to inform you that your manuscript, "Hematological and Immunological Profiles of Podoconiosis Patients in West Gojjam Zone, Ethiopia: A Comparative Cross-Sectional Study," has been formally accepted for publication in PLOS Neglected Tropical Diseases.

Best regards,

Shaden Kamhawi

co-Editor-in-Chief

Paul Brindley

co-Editor-in-Chief
